# Relatively Cyclic and Noncyclic *P*-Contractions in Locally 𝕂-Convex Space

**Edraoui Mohamed** [1] 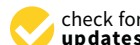**, Aamri Mohamed** [1] **and Lazaiz Samih** [2,*]

[1] Laboratory of Algebra, Analysis and Applications, Department of Mathematics, Ben M'sik Faculty of Sciences, University Hassan II, Casablanca 20000, Morocco
[2] Laboratory of Mathematical Analysis and Applications, Department of Mathematics, Dhar El Mahraz Faculty of Sciences, University Sidi Mohamed Ben Abdellah, Fes 30050, Morocco
[*] Correspondence: samih.lazaiz@gmail.com; Tel.: +212-062-2438-649

**Abstract:** Our main goal of this research is to present the theory of points for relatively cyclic and relatively relatively noncyclic *p*-contractions in complete locally 𝕂-convex spaces by providing basic conditions to ensure the existence and uniqueness of fixed points and best proximity points of the relatively cyclic and relatively noncyclic *p*-contractions map in locally 𝕂-convex spaces. The result of this paper is the extension and generalization of the main results of Kirk and A. Abkar.

**Keywords:** fixed point; locally 𝕂-convex spaces; relatively cyclic and relatively noncyclic *p*-contractions; best proximity point

## 1. Introduction

Let 𝕂 be a non-archimedean valued field, i.e., 𝕂 is neither ℝ nor ℂ, endowed with an absolute valued function $|.|$ such that

$$|x + y| \leq \max\{|x|, |y|\} \quad (x, y \in \mathbb{K})$$

Let $X$ be a topological vector space over 𝕂. A seminorm on the 𝕂-vector space $X$ is a map $p : X \to [0.\infty)$ satisfies

(i) $p(\lambda x) = |\lambda| \, p(x)$, $x \in X$ and $\lambda \in \mathbb{K}$.
(ii) $p(x + y) \leq \max\{p(x), p(y)\}$, $x, y \in X$

For a seminorm $p$ we have $p(0) = 0$ but $p(x)$ is allowed to be 0 for non-zero $x$. Note that each norm is a seminorm that vanishes only at 0.

Recall that a topological vector space $(X, \tau)$ over 𝕂 is called a (non-archimedean) locally 𝕂-convex space if $\tau$ has a basis of absolutely convex neighborhoods (a subset $A \subset X$ is called absolutely 𝕂-convex if $0 \in A$ and $ax + by \in A$ for all $x, y \in X$ and $a, b \in B_{\mathbb{K}}$ where $B_{\mathbb{K}} = \{a \in \mathbb{K} : |a| \leq 1\}$). Every locally 𝕂-convex topology can be generated in a natural way by some system of non-archimedean seminorms $\Gamma = \{p_\alpha\}$. A locally 𝕂-convex space $X$ is Hausdorff if and only if for each non-zero $x \in X$ there is a continuous seminorm $p$ on $X$ such that $p(x) \neq 0$. A sequence $\{a_1, a_2, \ldots\}$ in $X$ is called Cauchy net if and only if $\lim_n p(a_{n+1} - a_n) = 0$ for any seminorm $p$. This follows from

$$p(a_m - a_n) \leq \max\{p(a_m - a_{m-1}), \ldots, p(a_{n+1} - a_n)\}, \quad m > n.$$

A subset $S$ of a Hausdorff locally $\mathbb{K}$-convex space is called complete if each Cauchy net in $S$ converges to a limit that lies in $S$.For details, see [1–4].

On the other hand, the most fundamental fixed point theorem is the so-called Banach contraction principle (BCP for short), this result played an important role in various fields in mathematics. Due to its importance and simplicity, several authors have obtained many interesting extensions and generalizations of the Banach contraction principle. Ciric [5] introduced quasi-contraction map, which allowed him to generalize the Banach contraction principle.

In the absence of a fixed point, i.e., the equation $Tx = x$ has no solution, it is interesting to ask whether it is possible to find $(a, b) \in A \times B$ such that

$$p(a - Ta) = p(b - Tb) = D_p(A, B). \tag{1}$$

A point $\left(\bar{a}, \bar{b}\right) \in A \times B$ is said to be a best proximity pair for the mapping $T : A \cup B \to A \cup B$ if it is solution to the problem (1). Another interesting subject of the fixed point theory is the concept of cyclic contractions maps and the best points of proximity provided by Kirk et al. [6,7].

$(A; B)$ a nonempty pair of subsets of a locally $\mathbb{K}$-convex space $(X, \Gamma)$, we say that a mapping $T : A \cup B \to A \cup B$ is cyclic (resp. noncyclic) provided that $T(A) \subset B$ and $T(B) \subset A$ (resp. $T(A) \subset A$ and $T(B) \subset B$).

There are many results in this area see [8–12].

## 2. Fixed Point Results for Relatively Cyclic *P*-Contractions

In this section, we derive some fixed point theorems of certain relatively cyclic-type $p$-contractions in a complete locally $\mathbb{K}$-convex space.

**Definition 1.** *Let $A$ and $B$ be non empty subsets of locally $\mathbb{K}$-convex space $(X, \Gamma)$. A relatively cyclic map $T : A \cup B \to A \cup B$ is said to be relatively cyclic p-contraction if there exists $0 \leq \gamma_p < 1$ such that for all $p \in \Gamma$ and $a \in A$ and $b \in B$ we have*

$$p(Ta - Tb) \leq \gamma_p p(a - b). \tag{2}$$

**Theorem 1.** *Let $(X, \Gamma)$ be a complete Hausdorff locally $\mathbb{K}$-convex space, $A$ and $B$ be non empty closed subsets of $X$ and $T : A \cup B \to A \cup B$ a relatively cyclic p-contraction map. Then $T$ has a unique fixed point in $A \cap B$.*

**Proof.** Taking a point $a \in A$ since $T$ is $p$-contraction, we have

$$p\left(T^2 a - Ta\right) = p\left(T(Ta) - Ta\right) \leq \gamma_p p(Ta - a)$$

and

$$
\begin{aligned}
p\left(T^3 a - T^2 a\right) &= p\left(T\left(T^2 a\right) - T(Ta)\right) \\
&\leq \gamma_p\left(T^2 a - Ta\right) \\
&\leq \gamma_p^2 p(Ta - a)
\end{aligned}
$$

Inductively, using this process for all $n \in \mathbb{N}$ we have

$$p\left(T^{n+1} a - T^n a\right) \leq \gamma_p^n p(Ta - a)$$

Let $n \leq m$

$$
\begin{aligned}
p\left(T^m a - T^n a\right) &\leq \quad max\left\{p\left(T^m a - T^{m-1}a\right), p\left(T^{m-1}a - T^{m-2}a\right), ..., p\left(T^{n+1}a - T^n a\right)\right\} \\
&\leq \quad max\left\{\gamma_p^{m-1}p\left(Ta - a\right), \gamma_p^{m-2}p\left(Ta - a\right), .., \gamma_p^n p\left(Ta - a\right)\right\} \\
&\leq \quad \gamma_p^n p\left(Ta - a\right)
\end{aligned}
$$

Since $0 \leq \gamma_p < 1$, $\gamma_p^n \to 0$ as $n \to \infty$, we get $p\left(T^m a - T^n a\right) \to 0$, thus $\{T^n a\}$ is a $p$-Cauchy sequence. Since $(X, \Gamma)$ is complete, we have $\{T^n a\} \to \bar{a} \in X$. We note, that $\{T^{2n}a\}$ is a sequence in $A$ and $\{T^{2n-1}a\}$ is a sequence in $B$ in a way that both sequences tend to same limit $\bar{a}$. Since $A$ and $B$ are closed, we have that $\bar{a} \in A \cap B$. Hence $A \cap B \neq \emptyset$.

We claim that $T\bar{a} = \bar{a}$. Considering the condition relatively cyclic $p$-contraction we have

$$
\begin{aligned}
p\left(T^{2n}a - T\bar{a}\right) &= \quad p\left(TT^{2n-1}a - T\bar{a}\right) \\
&\leq \quad \gamma_p p\left(T^{2n-1}a - \bar{a}\right)
\end{aligned}
$$

Taking limit as $n \to \infty$ in above inequality, we have

$$
p\left(\bar{a} - T\bar{a}\right) \leq \gamma_p p\left(\bar{a} - T\bar{a}\right) < p\left(\bar{a} - T\bar{a}\right)
$$

This implies that $p\left(\bar{a} - T\bar{a}\right) = 0$. Since $X$ is Hausdorff, $T\bar{a} = \bar{a}$.

We shall prove that $\bar{a}$ is the existence of a unique fixed point of $T$. Clearly from (2) if $\bar{a}$ and $\bar{b}$ be two fixed points of $T$ we have

$$
p\left(\bar{a} - \bar{b}\right) = p\left(T\bar{a} - T\bar{b}\right) \leq \gamma_p p\left(\bar{a} - \bar{b}\right)
$$

Since $0 \leq \gamma_p < 1$ this implies $\bar{a} = \bar{b}$. Hence the proof is completed. $\square$

**Corollary 1.** *Let $A$ and $B$ be two non-empty closed subsets of a complete Hausdorff locally $\mathbb{K}$-convex space $X$. Let $T_1 : A \to B$ and $T_2 : B \to A$ be two functions such that*

$$
p\left(T_1\left(a\right) - T_2\left(b\right)\right) \leq \gamma_p p\left(a - b\right) \tag{3}
$$

*for all $p \in \Gamma$, $a \in A$ and $b \in B$ where $0 \leq \gamma_p < 1$. Then there exists a unique $\bar{a} \in A \cap B$ such that*

$$
T_1\left(\bar{a}\right) = T_2\left(\bar{a}\right) = \bar{a}
$$

**Proof.** Apply Theorem 1 to the mapping $T : A \cup B \to A \cup B$ defined by:

$$
T\left(a\right) = \begin{cases} T_1\left(a\right) & \text{if } a \in A \\ T_2\left(a\right) & \text{if } a \in B. \end{cases}
$$

Observe that condition (3) is reduced to condition (2). Then $T$ has a unique fixed $\bar{a} \in A \cap B$ such that

$$
T_1\left(\bar{a}\right) = T_2\left(\bar{a}\right) = \bar{a}.
$$

$\square$

**Theorem 2.** *Let $(X, \Gamma)$ be a complete Hausdorff locally $\mathbb{K}$-convex space, $A$ and $B$ two non empty closed subsets of $X$ and $T : A \cup B \to A \cup B$ be a relatively cyclic mapping that satisfies the condition*

$$
p\left(Ta - Tb\right) \leq \gamma_p max\left\{p\left(a - b\right), p\left(a - Ta\right), p\left(b - Tb\right)\right\} \tag{4}
$$

*for all $p \in \Gamma$, $a \in A$ and $b \in B$ and $0 \leq \gamma_p < 1$. Then, T has a unique fixed point in $A \cap B$.*

**Proof.** Let $a \in A$. By condition (4), we have

$$
\begin{aligned}
p\left(T^2 a - Ta\right) &= p\left(T\left(Ta\right) - Ta\right) \\
&\leq \gamma_p max\left\{p\left(Ta - a\right), p\left(Ta - T^2 a\right)\right\} \\
&\leq \gamma_p p\left(Ta - a\right).
\end{aligned}
$$

Similarly, we get $p\left(T^3 a - T^2 a\right) \leq \gamma_p^2 p\left(Ta - a\right)$.

Inductively, using this process for all $n \in \mathbb{N}$ we have

$$
p\left(T^{n+1} a - T^n a\right) \leq \gamma_p max\left\{p\left(Ta - a\right), p\left(Ta - T^2 a\right)\right\}
$$

thus

$$
\begin{aligned}
p\left(T^m a - T^n a\right) &\leq max\left\{p\left(T^m a - T^{m-1} a\right), p\left(T^{m-1} a - T^{m-2} a\right), ..., p\left(T^{n+1} a - T^n a\right)\right\} \\
&\leq max\left\{\gamma_p^{m-1} p\left(Ta - a\right), \gamma_p^{m-2} p\left(Ta - a\right), .., \gamma_p^n p\left(Ta - a\right)\right\} \\
&\leq \gamma_p^n p\left(Ta - a\right)
\end{aligned}
$$

Since $0 \leq \gamma_p < 1$, $\gamma_p^n \mapsto 0$ as $n \mapsto \infty$, we get $p\left(T^m a - T^n a\right) \to 0$. Hence $\{T^n a\}$ is a $p$-Cauchy sequence. As $(X, \Gamma)$ is complete, we have $\{T^n a\} \to \bar{a} \in X$. We note, that $\{T^{2n} a\}$ is a sequence in $A$ and $\{T^{2n-1} a\}$ is a sequence in $B$ so that the two sequences tend to the same limit $\bar{a}$. Since $A$ and $B$ are closed, we have that $\bar{a} \in A \cap B$ that is $A \cap B \neq \emptyset$.

Considering the condition (4) we have:

$$
\begin{aligned}
p\left(T^{2n} a - T\bar{a}\right) &= p\left(TT^{2n-1} a - T\bar{a}\right) \\
&\leq \gamma_p max\left\{p\left(T^{2n-1} a - \bar{a}\right), p\left(T^{2n-1} a - T^{2n} a\right), p\left(\bar{a} - T\bar{a}\right)\right\}
\end{aligned}
$$

Taking limit as $n \to \infty$ in above inequality, we have

$$
p\left(z - Tz\right) \leq \gamma_p p\left(z - Tz\right) < p\left(z - Tz\right)
$$

which implies that $p\left(\bar{a} - T\bar{a}\right) = 0$, since $X$ is Hausdorff, $T\bar{a} = \bar{a}$.

Clearly from (4) if $u$ and $v$ be fixed points of $T$ we have

$$
\begin{aligned}
p\left(u - v\right) &= p\left(Tu - Tv\right) \\
&\leq \gamma_p max\left\{p\left(u - v\right), p\left(u - Tu\right), p\left(v - Tv\right)\right\} \\
&\leq \gamma_p p\left(u - v\right)
\end{aligned}
$$

Since $0 \leq \gamma_p < 1$ this implies $u = v$. $\square$

**Corollary 2.** *Let $A$ and $B$ be two non-empty closed subsets of a complete Hausdorff locally $\mathbb{K}$-convex space $X$. let $T_1 : A \to B$ and $T_2 : B \to A$ be two functions such that*

$$
p\left(T_1\left(a\right) - T_2\left(b\right)\right) \leq \gamma_p max\left\{p\left(a - b\right), p\left(a - T_1\left(a\right)\right), p\left(b - T_2\left(b\right)\right)\right\} \tag{5}
$$

*for all $p \in \Gamma$ and $a \in A$ and $b \in B$ where $0 < \gamma_p < 1$. Then there exists a unique $\bar{a} \in A \cap B$ such that*

$$
T_1\left(\bar{a}\right) = T_2\left(\bar{a}\right) = \bar{a}
$$

**Proof.** Let $T : A \cup B \to A \cup B$ defined by

$$T(a) = \begin{cases} T_1(a) & \text{if } a \in A \\ T_2(a) & \text{if } a \in B \end{cases}$$

Then $T$ satisfies condition (4), we can now apply Theorem 2 to deduce that $T$ has a unique fixed point $\bar{a} \in A \cap B$ such that

$$T_1(\bar{a}) = T_2(\bar{a}) = \bar{a}$$

$\square$

## 3. Fixed Points of Relatively Noncyclic Mappings

In this section motivated by Theorem 3.1 [13], we prove the existence of a best proximity point of relatively noncyclic mappings and studied the existence of solution of problem (1) for relatively $p$-nonexpansive mappings in locally $\mathbb{K}$-convex.

**Definition 2.** *Let $(X, \Gamma)$ be a complete Hausdorff locally $\mathbb{K}$-convex space, $A$, $B \subset X$, we set*

$$\begin{aligned} A_0^p &= \{a \in A : p(a - b) = D_p(A, B), \text{ for some } b \in B\} \\ B_0^p &= \{a \in B : p(a - b) = D_p(A, B), \text{ for some } a \in A\} \end{aligned}$$

We extend the well known notion of $p$-property introduced in [5] for metric spaces to the case of locally $\mathbb{K}$-convex spaces.

**Definition 3.** *Let $(A, B)$ be a pair of nonempty subsets of a locally convex space $(X, \Gamma)$ with $A_0^p \neq \emptyset$. The pair $(A, B)$ is said to have $p$-property iff*

$$\begin{cases} p(a_1 - b_1) = D_p(A, B) \\ p(a_2 - b_2) = D_p(A, B) \end{cases} \implies p(a_1 - a_2) = p(b_1 - b_2) \quad (\forall p \in \Gamma).$$

*where $a_1, a_2 \in A_0^p$ and $b_1, b_2 \in B_0^p$*

**Definition 4.** *Let $(A, B)$ be a pair of nonempty subsets of a locally convex space $(X, \Gamma)$. A mapping $T : A \cup B \to A \cup B$ is called relatively $p$-nonexpansive iff $p(Ta - Tb) \leq p(a - b)$ for all $p \in \Gamma$ and $(a, b) \in A \times B$. If $A = B$, we say that $T$ is $p$-nonexpansive.*

**Lemma 1.** *[14] Let $(X, \Gamma)$ be a complete Hausdorff locally $\mathbb{K}$-convex space if $T : X \to X$ is a $p$-contraction mapping then $T$ has a unique fixed point $\bar{x}$ in $X$, and $T^k x \to \bar{x}$ for every $x \in X$.*

**Proof.** Let $y \in X$ and $k \geq 1$ we have

$$\begin{aligned} p\left(T^k y - y\right) &\leq \max\left\{p\left(T^k y - T^{k-1} y\right), p\left(T^{k-1} y - T^{k-2} y\right), .., p(Ty - y)\right\} \\ &\leq \max\left\{\gamma^k p(Ty - y), \gamma^{k-1} p(Ty - y), .., p(Ty - y)\right\} \end{aligned}$$

then $\max\left\{\gamma^k p(Ty - y), \gamma^{k-1} p(Ty - y), .., p(Ty - y)\right\} = p(Ty - y)$, which implies that for all $x \in X$ and $k \geq 1$

$$p\left(T^k x - x\right) \le p\left(Tx - x\right).$$

For every $p \in \Gamma$ and $k \ge 1$, Choose $n$ sufficiently large. Then for $y = T^n x$, we have

$$
\begin{aligned}
p\left(T^{n+k}x - T^n x\right) &\le& p\left(T^{n+1}x - T^n x\right) \\
&\le& \gamma_p^n p\left(Tx - x\right)
\end{aligned}
$$

Since $0 \le \gamma_p < 1$, $\gamma_p^n \to 0$ as $n \to \infty$, we get $p\left(T^{n+k}x - T^n x\right) \to 0$. Thus $\left\{T^k x\right\}$ is a $p$-Cauchy sequence and so it converges to a point $\overline{x}$ in $X$. Clearly $T\overline{x} = \overline{x}$ and uniqueness of the fixed point follows as usual since $X$ is Hausdorff. □

**Theorem 3.** *Let $(X, \Gamma)$ be a complete Hausdorff locally $\mathbb{K}$-convex space and $(A, B)$ be two nonempty closed subsets of $X$. Assume that $T : A \cup B \to A \cup B$ is a relatively noncyclic mapping such that for some $\gamma_p \in (0, 1)$*

$$p\left(Tx - Ty\right) \le \gamma_p p\left(a - b\right)$$

*for all $p \in \Gamma$ and $(a, b) \in A \times B$ then $D_p\left(A, B\right) = 0$. Moreover, the mapping $T$ has a fixed point in $A \cup B$ if and only if $A \cap B \ne \emptyset$.*

**Proof.** Let $\{a_n\}$ and $\{b_n\}$ be two sequences in $A$ and $B$ respectively such that $p\left(a_n - b_n\right) \to D_p\left(A, B\right)$. Then

$$D_p\left(A, B\right) \le p\left(Ta_n - Tb_n\right) \le \gamma_p p\left(a_n - b_n\right).$$

Taking limit when $n$ tends to infinity, we see that necessarily $D_p\left(A, B\right) = 0$. Suppose first that $A \cap B \ne \emptyset$. If we apply the Theorem 1 in $A \cap B$, there exists a fixed point of $T$ that in fact is unique in $A \cap B$.

On the other hand, suppose that $T$ has a fixed point $\overline{b}$ in $A \cup B$. Without loss of generality, suppose that $\overline{b} \in B$. Then, given a point $a_0 \in A$, if we denote $a_n = T^n a_0$ we have

$$p\left(a_n - \overline{b}\right) \le \gamma_p p\left(a_{n-1} - \overline{b}\right) \le \gamma_p^2 p\left(a_{n-2} - \overline{b}\right) \le \cdots \le \gamma_p^n p\left(a_0 - \overline{b}\right)$$

Since $0 \le \gamma_p < 1$, $\gamma_p^n \to 0$ as $n \to \infty$, we get that $\{a_n\}$ converges to $\overline{b}$. Since $A$ is closed, $\overline{a} \in A \cap B$ and the result follows. □

**Theorem 4.** *Let $(X, \Gamma)$ be a complete Hausdorff locally $\mathbb{K}$-convex space and $(A, B)$ be two nonempty closed subsets of $X$ such that $A_0^p \ne \emptyset$. Assume that $(A, B)$ satisfies the p-property. Let $T : A \cup B \to A \cup B$ be a relatively relatively noncyclic mapping that satisfies the conditions*

(i) *$T_{|A}$ is p-contraction,*
(ii) *$T$ is relatively p-nonexpansive.*

*Then the minimization problem (1) has a solution*

**Proof.** Let $a \in A_0^p$ then exists $b \in B$ such that $p\left(a - b\right) = D_p\left(A, B\right)$. Since $T$ is relatively $p$-nonexpansive; so

$$p\left(Ta - Tb\right) \le p\left(a - b\right) = D_p\left(A, B\right)$$

Hence, $Ta \in A_0^p$, therefore $T\left(A_0^p\right) \subseteq A_0^p$. Now let $a_0 \in A_0^p$. By Lemma 1 if $a_{n+1} = Ta_n$, then $a_n \to \bar{a}$ where $\bar{a}$ is a fixed point of $T$ in $A$. Since $a_0 \in A_0^p$, then exists $b_0 \in B$ such that $p\left(a_0 - b_0\right) = D_p\left(A, B\right)$. Again, since $a_1 = Ta_0 \in A_0^p$, then there exists $b_1 \in B$ such that $p\left(a_1 - b_1\right) = D_p\left(A, B\right)$.

Inductively, using this process for all $n \in \mathbb{N} \cup \{0\}$ we have a sequence $\{b_n\}$ in $B$ such that

$$p\left(a_n - b_n\right) = D_p\left(A, B\right).$$

Since $(A, B)$ has the $p$-property, we get that for all $n, m \in \mathbb{N} \cup \{0\}$

$$p\left(a_n - b_m\right) = p\left(a_n - b_m\right).$$

This implies that $\{b_n\}$ is a Cauchy sequence, and hence there exists $\bar{b} \in B$ such that $a_n \to \bar{b}$. We now have

$$p\left(\bar{a} - \bar{b}\right) = \lim_{n \to \infty} p\left(a_n - b_n\right) = D_p\left(A, B\right)$$

We know that $T$ is relatively nonexpansive, so that

$$p\left(T\bar{a} - T\bar{b}\right) \leq p\left(\bar{a} - \bar{b}\right) = D_p\left(A, B\right)$$

Thus $p\left(\bar{a} - T\bar{b}\right) = p\left(\bar{a} - T\bar{b}\right)$, since $(A,B)$ has property $P$. Hence $\left(\bar{a} - \bar{b}\right) \in A \times B$ is a solution of (1). $\quad\square$

**Author Contributions:** Conceptualization, E.M.; Supervision, A.M. and L.S.; Validation, A.M.; Writing—original draft, T.S. and A.B.

**Funding:** This research received no external funding.

**Conflicts of Interest:** Research was supported by a National Centre of Scientific and Technological Research grant. The authors would like to express their gratitude to the editor and the anonymous referees for their constructive comments and suggestions, which have improved the quality of the manuscript..

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
