# Peer review of "Relatively Cyclic and Noncyclic P-Contractions in Locally K-Convex Space"

_axioms, doi:10.3390/axioms8030096_

Round 1

Reviewer 1 Report

I have read whole paper. I cannot recommend it for publication without
any example which support theoretical result. Hence, let authors add at
least two examples as the support.

Author Response

 Hello Sir, Thank you for considering me as a reviewer for this publication in your esteemed journal

Thank you for your note

The concept of the fixed point in locally K convex space is new

Sir told you that we could not find any examples, given the lack of references

Sir told you that we could not find any examples, given the lack of references

Reviewer 2 Report

In this paper, the authors studied the theory of relatively cyclic and noncyclic P-contractions In Locally K-Convex Space. The paper seems interesting, but there are some questions should be addressed.   

1. The authors state that the result in this paper is the extension and generalization of the main results of Kirk and A. Abkar.

Please explain this slightly where appropriate. And by the way, please also give the motivation of this paper.

2.The references seems a little old, no recent work related to the paper?

3.The layout of the paper should be improved. There are many punctuation and printing problems.

(1) Abstract. Line 5, should the word ``. "Global be deleted?

(2) Many periods and commas are missing. Please add them.

(3) Some words should not be indented. For example,  page 1, line 15,``for all.

(4) Page 2, line 17, Cauchy net or Cauchy sequence?

(5) Some Spaces are missing. For example, page 2, line 36-37, before the word ``and

(6) Definition 1.

   (i) Please remove the excess n.

   (ii) It shouldn't be a newline.

(7) Page 6, line 97, thereforev should be therefore.

(8) Page 7, line 100, (A,B) should be should be italicized.

Recommendation: accepted after revision.

Author Response

 Hello Sir: Thank you for considering me as a reviewer for this publication in your esteemed journal

All errors have been treated

Thanks for the recommendations

cordially

Reviewer 3 Report

In this paper, the authors presented the theory of points for relatively cyclic and 2 relatively relatively noncyclic p-contractions in complete locally K−convex spaces by providing basic 3 conditions to ensure the existence and uniqueness of fixed points and best proximity points of the 4 relatively cyclic and relatively noncyclic p-contractions map in locally K−convex spaces . The main results are interesting. I'd like to recommend it for publication. However, the following points should be revised. (1) Introduction should be reorganized. Motivation, background and development should be stated in Section 1. (2) An example is needed to support their results. (3) To avoid the possible conflict and further support this manuscript, the following very related references should be included: [A.J. Zaslavski, Two fixed point results for a class of mappings of contractive type, J. Nonlinear Var. Anal. 2 (2018), 113-119], [I. Altun, G. Durmaz, M. Olgun, P-contractive mappings on metric spaces, J. Nonlinear Funct. Anal. 2018 (2018), Article ID 43], [X. Qin, J.C. Yao, Projection splitting algorithms for nonself operators, J. Nonlinear Convex Anal. 18 (2017), 925–935] and [Sehie Park, Some general fixed point theorems on topological vector spaces, Appl. Set-Valued Anal. Optim. 1 (2019), 19-28]. (4) A concluding remark is needed to summarize this manuscript. (5) There are some typos and English presentation also needs to be improved.

Author Response

 hello 

Thank you for considering me as a reviewer for this publication in your esteemed journal

An article has been processed and recommended references have been added

Thank you for your support and your 

cordially

Round 2

Reviewer 1 Report

The report of the paper: Relatively Cyclic And Noncyclic P-Contractions In Locally K-Convex Space

Dear from the journal, dear editor

I have consider new version carefully and I can say the next:

Authors accepted all remarks for previous version. I am sure that this new version is complete (indeed) suitable for publication in your esteemed journal.

Hence, I strongly recommend it for publication without any new improvements.